e-science/complexity

social networks, corruption, social capital

**Author for correspondence:**
Johannes Wachs
e-mail: johanneswachs@gmail.com

# Social capital predicts corruption risk in towns

Johannes Wachs[1], Taha Yasseri[2,3], Balázs Lengyel[4,5] and János Kertész[1,6]

[1]Department of Network and Data Science, Central European University, Budapest 1051, Hungary
[2]Oxford Internet Institute, University of Oxford, 1 St Giles, Oxford OX1 3JS, UK
[3]Alan Turing Institute, 96 Euston Road, London NW1 2DB, UK
[4]Agglomeration and Social Networks Lendület Research Group, Hungarian Academy of Sciences, Budapest 1097, Hungary
[5]International Business School Budapest, Budapest 1037, Hungary
[6]Institute of Physics, Budapest University of Technology and Economics, Budapest 1111, Hungary

JW, 0000-0002-9044-2018; TY, 0000-0002-1800-6094

Corruption is a social plague: gains accrue to small groups, while its costs are borne by everyone. Significant variation in its level between and within countries suggests a relationship between social structure and the prevalence of corruption, yet, large-scale empirical studies thereof have been missing due to lack of data. In this paper, we relate the structural characteristics of social capital of settlements with corruption in their local governments. Using datasets from Hungary, we quantify corruption risk by suppressed competition and lack of transparency in the settlement's awarded public contracts. We characterize social capital using social network data from a popular online platform. Controlling for social, economic and political factors, we find that settlements with fragmented social networks, indicating an excess of *bonding social capital* has higher corruption risk, and settlements with more diverse external connectivity, suggesting a surplus of *bridging social capital* is less exposed to corruption. We interpret fragmentation as fostering in-group favouritism and conformity, which increase corruption, while diversity facilitates impartiality in public life and stifles corruption.

## 1. Introduction

Corruption is widely recognized to affect adversely social and economic outcomes of societies [1], yet it is difficult to fight [2]. Though education and income seem to decrease corruption [3], it persists even under highly developed, democratic conditions and has significant regional variation within countries [4]. Researchers often relate corruption to social aspects of society such as segregation [5], interpersonal trust [6], civic-mindedness [7] and

community engagement [8]. These approaches build on the insight that corruption is a collective outcome of a community shaped by the interactions among individuals [9], suggesting that differences in social capital, especially in the network structure, may help explain the persistence of corruption and the observed differences in its levels.

The concept of social capital or the 'connections among individuals—social networks and the norms of reciprocity and trustworthiness that arise from them' [8] is usually applied to understand the behaviour of individuals [10]. Yet, city- or country-level aggregations have also proved useful [8], for example in studying economic development and prosperity [11]. As a communal quantity, social capital is a sort of public good embedded in a social network [12] of a settlement. Given the aforementioned relationship between corruption and social capital, it is therefore, as suggested above, natural to expect that the structure of social capital at the settlement level has considerable impact on the scale of corruption in that community. Despite a significant interest in the network aspects of corruption [13] and recent experimental evidence that corruption has collaborative roots [14], less is known about how the patterns of connectivity of a whole society influence the general level of corruption in its government.

Previous studies of relating social capital and corruption [15,16] have been constrained by two empirical challenges: the difficulty of measuring corruption and the lack of data on network structure at the settlement level. Corruption is one of the most hidden type of crimes; therefore, it is difficult to estimate its extent in general, even with significant local information. For example, an audit study of corruption in rural Indonesia road construction finds that, independent of objective measures of corruption risk, villager perceptions of corruption are significantly distorted by factors such as ethnic diversity [17].

Many studies measure corruption using national or regional surveys [2] and suffer from the subjectivity of corruption perceptions [18]. Other studies use data on the frequency of investigations and convictions of politicians [3], in which a source of bias may be that in places where corruption is prevalent the judiciary is more likely to be corrupt itself [19]. Recent efforts to clean and standardize large datasets on public procurement [20] have been very helpful in this context as their study can lead to new, more objective indicators of corruption risk.

In the absence of direct network data, researchers often quantify social capital using proxies such as rates of voting, donating blood and volunteering [7]. As these rates are themselves related to the underlying social networks, they indicate the relevance of social capital and trust instead of explaining the causes of corruption in terms of network structure. Mapping out the social capital at the level of settlements using traditional tools is a formidable task. Fortunately, recent developments in information-communication technologies and their increasing popularity present large datasets containing relevant information. For example, data from online social networks and cellphone records have been used to relate connectivity and socio-economic outcomes [21–25].

In this paper, we propose to characterize the level of corruption risk in settlements in terms of their social capital using two sources of micro-level data from Hungary. We quantify the structural characteristics of settlements' social capital using complete data from 'iWiW', a now defunct online social network once used by approximately 40% of the adult Hungarian population [26]. We measure corruption risk using administrative data on public procurement contracts over a period of 8 years [27].

Public procurement contracts constitute a major channel of public funds to private hands and are highly vulnerable to corruption [20]. Recently, a set of corruption risk indicators have been derived from public contract data, for example, counting how often contracts attract only a single bidder. Averaged to the regional or national levels, these contract-based corruption risk measures have been shown to correlate with corruption perception surveys [20], quality of government indicators [28], and higher cost outcomes for internationally comparable goods such as CT/CAT scanners [29]. In the Hungarian case, we find that settlements involved in a recent corruption scandal [30] have significantly higher corruption risk in their contracts.

Putnam distinguishes between two structural categories of social capital: *bonding* and *bridging* social capital [8], and we expect that these have different impacts on corruption risk. Bonding social capital is based on the phenomenon of closure in a social network, describing the extent to which people form dense, homogeneous communities. Such communities have benefits: members share high levels of trust and can count on each other in times of crisis. They can also be confident that members who defy the norms of the community will be censured [31]. The homogeneity of such tight-knit communities is often based on ethnicity, religion or class [32], indicating possible drawbacks to bonding social capital: homogeneity facilitates conformity and implies exclusion of outsiders [33]. Solidarity can reach the extent that insiders will protect each other even if norms from a wider context are broken, in some cases even if crimes are committed. Sophisticated criminal organizations like the Mafia, members of which may regularly be faced with great incentives to 'flip', rely on bonding rituals, ethnic homogeneity, and family

ties to enforce solidarity and in-group trust [13,34]. The negative effects of excessive bonding social capital on society are not limited to crime and corruption. Entrepreneurs embedded in dense networks are disadvantaged because of pressure to employ under-qualified relatives [35], while ethnically homogeneous groups of traders are more likely to overprice financial assets held by their co-ethnics, causing financial bubbles to form [36].

Bridging social capital, on the other hand, refers to the connections between people from different social groups. Such ties are valuable for their ability to convey novel information [37] and exposure to diverse perspectives, though they do not serve as reliable sources of support in hard times. Previous work shows, for instance, that immigrants in The Netherlands with bridging connections outside their ethnic group have significantly higher incomes and employment rates [38]. But, bridging social capital is not only thought to be useful for the resources it allocates. Using an agent-based model, Macy and Skvoretz showed how trust emerged among densely connected neighbours and diffused in a social network via weak ties [39], implying that low bridging social capital restricted trust to within-group interactions. Indeed, empirical evidence showed that ethnic groups in diverse communities with more bridging social capital evaluate each other more positively [40].

The two concepts of bonding and bridging social capital exist in tension with each other. They reflect, to quote Portes, 'Durkheim's distinction between mechanical solidarity, based on social homogeneity and tight personal bonds, and organic solidarity, based on role differentiation, impersonal norms, and an extensive division of labor' [41]. A settlement in which mutual cooperation relies excessively on mechanical solidarity will tend towards norms of in-group favouritism or particularism [2]. Individuals in such a society will tend to make choices, for example, in the allocation of public resources, distinguishing between insiders and outsiders based on a feeling of security rather than trust [42]. By contrast, when cooperation is built on impersonality, general trust facilitates impartial outcomes.

We therefore pose two hypotheses relating bonding and bridging social capital to local corruption risk. The first (H1) is that excess bonding social capital, indicating the potential presence of norms of in-group favouritism in a settlement is correlated with higher corruption risk in its government. The second (H2) is that a high level of bridging capital, including connections to other settlements, is correlated with lower levels of corruption risk because it fosters impersonal and universalistic norms. Where mechanical solidarity or bonding social capital dominates organic solidarity or bridging social capital, universalistic norms under which public markets are thought to function best are unsustainable. These hypotheses suggest why corruption is so difficult to fight: it is embedded in the social network of a place.

Previous work using survey data is in accord with our hypotheses. Harris finds a significant positive relationship between excess bonding social capital, measured using surveys, and corruption across over 200 countries [16]. In a comparative study of the 50 USA states, Knack finds that residents in states with higher census response and volunteering rate their governments' performances more highly [43]. He finds no such effect for rates of membership in social clubs, a more exclusive form of socialization than volunteering. Paccagnella & Sestito [44] find that in regions with high electoral turnout and blood donation rates, Italian schoolchildren cheat less frequently on standardized tests. In schools with greater ethnic homogeneity and with hometown teachers, cheating is more frequent. These case studies and indirect evidence give some support the above hypotheses; however, there is need for studies based on more direct data at multiple levels.

We find significant evidence for our hypotheses using multivariate regression models to relate corruption risk and structural aspects of social capital. Hungarian settlements with fragmented social networks, which we interpret as evidence of excess bonding social capital, have higher corruption risk in their public procurement contracts. But, if the typical resident of a settlement has more diverse connections, especially over the boundaries of their own settlements, then local corruption risk is lower. These results hold controlling for several potential confounders including economic prosperity, education, demographics and political competitiveness.

# 2. Empirical setting and methods

## 2.1. Public contracting

In OECD economies, procurement typically accounts for between 10 and 20% of GDP [45] covering everything from school lunches to hospital beds and highway construction. The complexity of the

**Table 1.** Elementary indicators of public contract corruption risk. More detail is provided in the electronic supplementary material.

| indicator and symbol | values | indicator definition |
|---|---|---|
| single bidder $C_{singlebid}$ | {0, 1} | 1 if a single firm submits an offer. |
| closed procedure $C_{closedproc}$ | {0, 1} | 1 if the contract was awarded directly to a firm or by invite-only competition. |
| no call for bids $C_{nocall}$ | {0, 1} | 1 if no call for bids was published in the official procurement journal. |
| long eligibility criteria $C_{eligcrit}$ | {0, 1} | 1 if the length in characters of the eligibility criteria for firms to participate in the tender is above the market average.[a] |
| extreme decision period $C_{decidetime}$ | {0, 1} | 1 if the award was made within 5 days of the deadline or more than 100 days following. |
| short time to submit bids $C_{bidtime}$ | {0, 0.5, 1} | 1 if the number of days between the call and submission deadline is less than 5, 0.5 if between 5 and 15. |
| non-price criteria $C_{nonprice}$ | {0, 1} | 1 if non-price criteria are used to evaluate bids. |
| call for bids modified $C_{callmod}$ | {0, 1} | 1 if the call for bids was modified. |

[a]We define a market in terms of two-digit common procurement vocabulary (CPV) codes, an EU-wide taxonomy of goods and services [48].

contracts and the relative inelasticity of the government's demand for goods make them a prime target for corruption [46].

Contracts are supposed to be awarded using impartial market mechanisms [47]: open and fair competition for a contract is considered the best way to ensure that the government makes purchases of good quality at the lowest cost. Usually, an issuer of a contract publishes a call for bids from the private sector, setting a deadline for submissions leaving enough time for broad participation. Companies submit sealed offers, including a price. The company offering to provide the good or service for the lowest price, meeting the standards set in the call for bids, wins the contract.

### 2.1.1. Measuring settlement corruption risk in contracting

Corruption in public contracting typically involves the restriction of competition. If corrupt bureaucrats wish to award a contract to a favoured firm, they must somehow exclude other firms from participating in the competition for the contract. We quantify this phenomenon at the contract level by tracking the presence of elementary corruption indicators, signals we can extract from metadata suggesting that competition may have been curbed [20]. These quantitative indicators [27], deduced from qualitative work on corruption in public contracting, are the fingerprints of techniques used to steer contracts towards preferred firms. We consider eight such elementary indicators, defined in table 1. From these eight elementary indicators, we define two measures of contract corruption risk:

*Closed procedure or single bidding* ($C_{csb}$). Did the contract attract only a single bid **or** was the contract awarded by some procedure besides an open call for bids, for example by direct negotiation with a firm or by an invitation-only auction? In terms of the indicators defined above:

$$C_{csb} = \max(C_{singlebid}, C_{closedproc})$$

*Corruption risk index* (CRI). Following [27], we average all eight elementary indicators defined in table 1 for each contract:

$$\mathrm{CRI} = \frac{1}{8}(C_{singlebid} + C_{closedproc} + C_{nocall} + C_{eligcrit} + C_{bidtime} + C_{nonprice} + C_{callmod} + C_{decidetime}).$$

These indicator-based measures of corruption risk have been related to traditional measures of corruption at the regional and national levels. Among EU countries, similar indicators are correlated ($\rho \approx 0.5$) with both the World Bank's Control of Corruption rankings and Transparency International's Corruption Perceptions Index [20]. We propose that our indicators supplement these perception-based measures with more objective data at a micro-scale.

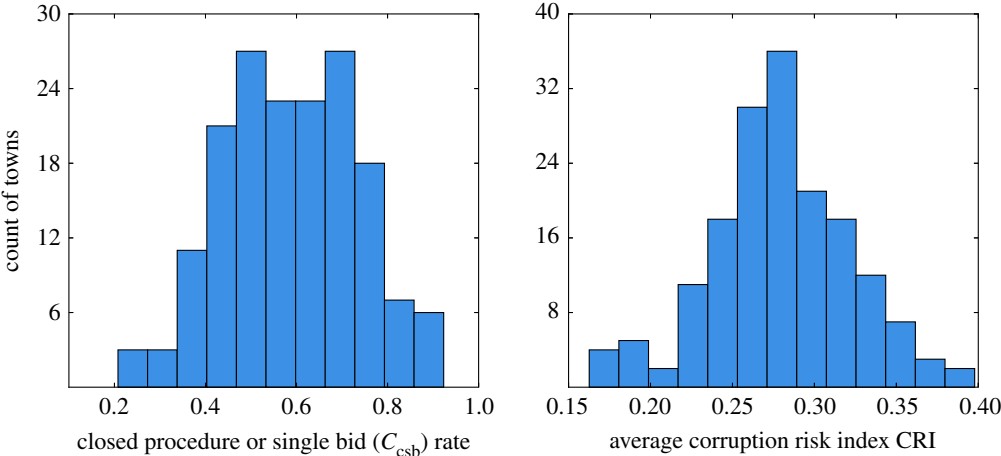

**Figure 1.** Distributions of average contract corruption risk indicators across Hungarian settlements.

Our indicators also predict cost overruns and price inflation in European infrastructure projects [29]. At the micro-level, public bodies issuing high corruption risk contracts are significantly more likely to award contracts to new companies after a change in government [49]. Finally, evidence from the USA suggests that firms making campaign contributions are awarded contracts with higher corruption risk [50].

### 2.1.2. Local government contracting data

We examine 20 524 municipal government contracts from the period of 2006–2014 issued by Hungarian settlements awarding at least five contracts a year on average. We exclude towns issuing fewer contracts because we are interested in systematic patterns of corruption over a sustained period of time. Our indicators applied to individual contracts are only noisy measures of corruption—it is rather the consistent observation of red flags in contracting over time that suggests that a town has a significant problem with persistent corruption risk. Our results are robust to including towns issuing at least one contract per year on average, reported in the electronic supplementary material.

Our goal is to quantify the overall level of corruption risk in a settlement over the full period for which we have data. We create two such scores by averaging the risk indicators defined above over all contracts issued by the settlement. We arrive at two measures of settlement corruption risk: the rate at which a settlement issued closed-procedure or single bid contracts ($C_{csb}$), and the average corruption risk index (CRI) score of its contracts.

There are 169 settlements in Hungary meeting the minimum contracting criterion, excluding Budapest. We exclude Budapest for two reasons: because it is a severe outlier in size and economic importance and because of its unique governance structure. Budapest is split into 23 districts, each with its own local government and mayor. It also has a city-wide government and mayor. As iWiW treats Budapest as a single settlement and as many contracting decisions are taken at the district level, we judged that we could not reasonably compare the full city with other settlements in Hungary.

We plot the distributions of the settlement corruption risk scores in figure 1. We note that there is substantial variation across settlements: some award over 90% of their contracts either via a closed procedure or to a single bidding supplier, while others do so less than 25% of the time.

As a test of the validity of our settlement-level measures of corruption risk, we check them against a near-ground truth case of corruption. In 2018, OLAF, the European anti-fraud agency reported that 35 Hungarian local government public lighting contracts awarded between 2010 and 2014 contained 'serious irregularities' [30]. Elios, the company winning these contracts, was owned at that time by the son-in-law of the Hungarian Prime Minister. The contracts are considered to be overpriced and the Hungarian government was appealed for initiating an investigation, which has already started.

These cases provide a useful test of our corruption risk indicators. There is compelling evidence that settlements implicated in the scandal have, at least once, rigged a public procurement contract to favour a connected firm. We compare the average corruption risk indicators of the 35 settlements that awarded lighting contracts to Elios in the period in question with all other settlements in our sample in figure 2. Using a Mann–Whitney U-test, we find that settlements involved in the scandal have

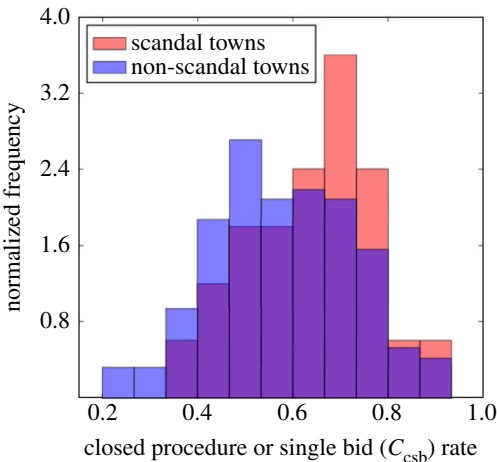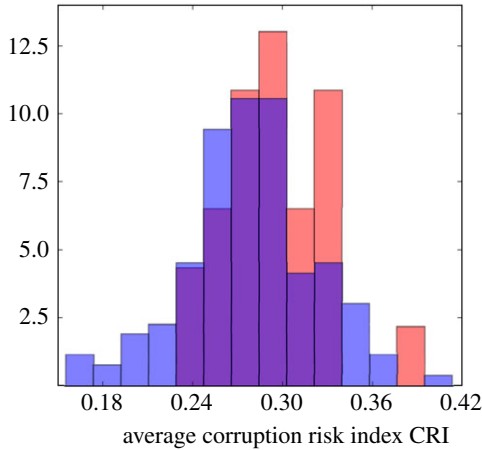

**Figure 2.** Distributions of average contract corruption risk indicators for settlements involved in the Elios scandal compared with all other settlements. Settlements involved in the scandal have significantly higher average corruption risk in their contracting than their counterparts.

significantly higher rates of corruption risk according to both measures (64% versus 58% $C_{csb}$ rate, $U = 1385$, $p = 0.033$; 0.30 versus 0.28 average CRI, $U = 1397$, $p = 0.037$).

## 2.2. Measuring social capital

iWiW was a popular online social network operating in Hungary from 2004 to 2013. At its peak, it boasted over 3.5 million active users (out of a population of around 10 million) and was among the top 3 most visited sites in the country. After a period of sustained popularity, it finally collapsed in 2013 as competitors, including Facebook, conquered the market. The increasing tendency of users to leave led to cascades in the social network, highlighting the networked nature of the site [51,52]. Geographical proximity is a major positive predictor of the likelihood of friendship ties on iWiW, and connections between settlements reflect historical administrative boundaries and geographical barriers [26].

The iWiW network consists of users as nodes and mutually acknowledged friendship ties between users as links. Data from iWiW include information on each user's settlement, selected from a menu. We used an anonymized version of the data to ensure privacy. We consider all nodes and links in the network present at the end of 2012, during the peak of its use and before the most significant period of turnover on the site leading to its collapse, to define our measures of bonding and bridging social capital. We consider this aggregated network rather than an evolving network from year to year because we do not have repeated interactions in our data (in contrast with the repeated links that can be observed in cellular phone call data) and because we are interested in a long-run characterization of the social network structure of settlements. We describe steps we took to clean the data and the distribution of user rates at the settlement level in the electronic supplementary material.

Despite valid concerns about the representativity of data taken from online social networks [53], studies indicate that data from online social networks offer a useful picture of the social capital of their users [54,55]. As adoption of online social networks increases, they become increasingly useful for the study of the social structures [56]. In any case, we control for possible confounding factors including settlement average income, rate of iWiW use and share of the population over 60 in our models.

### 2.2.1. Fragmentation

Our first settlement-level network measure, *fragmentation*, quantifies the extent to which people in the settlement form densely connected and well-separated communities. We do not consider the links residents of a settlement have with other settlements. Fragmentation measures a settlement's bonding social capital. Before we proceed, we note that 'settlement' will always be used to refer to a municipality, while 'community' refers to a group of nodes detected in the iWiW social network of a settlement using a network science algorithm, in other words a subset of the nodes of the town which are densely connected.

We measure fragmentation of the settlement's internal social network using a community detection method to identify communities of highly connected nodes. We use the Louvain algorithm [57], a popular and efficient method leading to a partition of the network. We measure the quality of the partition, the tendency of edges to be within rather than between the detected communities, using modularity [58]. Given a social network of users in a settlement $S$ and a partition of the network's nodes into $K$ communities, the modularity $Q(S)$ of the partition of the network can be written as follows:

$$Q(S) = \sum_{k=1}^{K} \left[ \frac{L_k^w}{L} - \left( \frac{L_k}{L} \right)^2 \right],$$

where $L$ is the total number of edges in the considered network, $L_k$ is the number of edges adjacent to members of community $k$ and $L_k^w$ is the number of edges within community $k$.

As modularity is highly dependent on the size and density of the network [59], we scale each settlement's modularity score in order to make valid comparisons between the settlements. Following Sah *et al.* [60], we divide each settlement's modularity score by the theoretical maximum modularity $Q_{max}(S)$ that the given partition could achieve, namely if all edges were within communities.

$$Q_{max}(S) = \sum_{k=1}^{K} \left[ \frac{L_k}{L} - \left( \frac{L_k}{L} \right)^2 \right].$$

We then define the *fragmentation* $F_S$ of a settlement $S$ as the quotient

$$F_S = \frac{Q(S)}{Q_{max}(S)}. \tag{2.1}$$

Fragmentation measures the tendency of individuals to belong to distinct communities within a settlement. A fragmented settlement consists of tightly knit communities that are weakly connected. Both the excess of connections within and the rarity of connections between communities in fragmented networks are relevant to our theoretical framing of the origins of corruption as they indicate excess bonding social capital. The high density of connections within a community facilitates the enforcement of reciprocity, while having few connections between communities fosters particularism.

To better understand the concept of fragmentation, we compare two settlements, one at the 90th percentile of fragmentation (settlement *a*) and the other at the 10th percentile (settlement *b*). The two settlements have populations of roughly 10 000 and have iWiW user rates between 30 and 35%. We randomly sample 300 users from both social networks for the sake of visualization and plot their connections in figure 3. Settlement *a* is clearly more fragmented than settlement *b*. We also show the full adjacency matrices of the networks of these settlements, grouping nodes by their detected communities into blocks on the diagonal shaded in red. We label each community by the share of its edges staying within the community. The fragmented settlement has a clear over-representation of within-community edges.

## 2.2.2. Diversity

Past research on online social networks noted that users connect with people from a variety of focal experiences in their life-course [54]. For example, a user may connect with her schoolmates, university classmates, coworkers, family, and friends from environments including social clubs, sports teams or religious communities. We measure the diversity of an individual user's network by the (lack of) overlap between these foci. In this case, we do not restrict our attention to edges between users from the same settlement.

Following Brooks *et al.* [54], we consider the connections among the friends of a focal user or ego, i.e. the ego network without the ego. We then detect communities in the resulting network using the Louvain algorithm [57]. We can assume that members of a community of alters share some common context. We measure the separation of these communities of alters using modularity. Low modularity indicates that a user's connections tend to know each other, and that the user's different spheres of life involve the same people. High modularity indicates that the ego has a bridging role between weakly connected communities, and so we refer to such users as having high diversity in their social networks. We show examples of low and high diversity users with networks of similar sizes in figure 4.

We aggregate this user-level measure to a measure of settlement diversity $D_S$ by averaging each user's modularity score

$$D_S = \frac{1}{|S|} \sum_{i \in S} Q(\{alters_i\}),$$

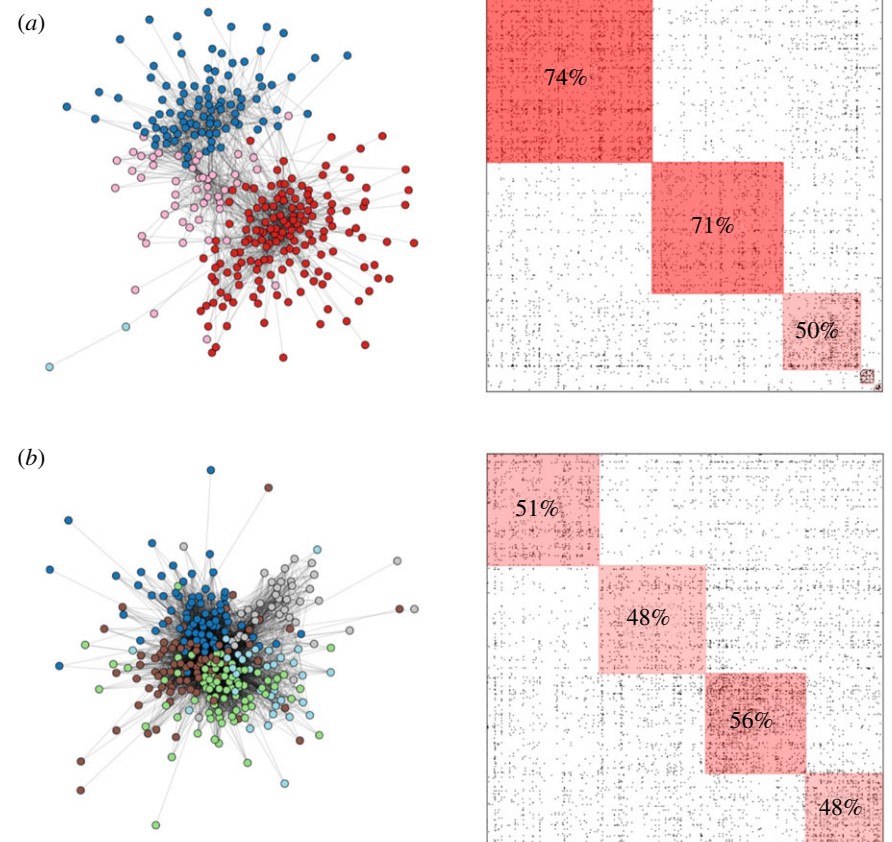

**Figure 3.** Sampled social networks and adjacency matrices of high (*a*) and low (*b*) fragmentation settlements. Node colours indicate membership in communities. In the adjacency matrices, percentages indicate the share edges staying within each community. In the fragmented settlement, communities have significantly fewer connections with other communities.

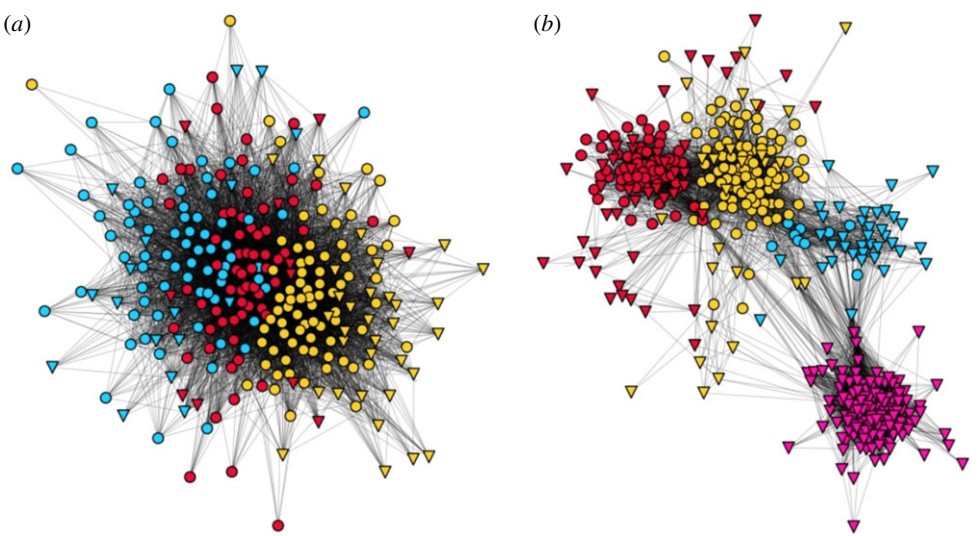

**Figure 4.** Ego networks with low (*a*) and high (*b*) diversity. Colours indicate membership in detected communities in the ego network. Circles denote users from the same settlement as the ego, while triangles mark users from elsewhere. The high diversity user's network has clusters of alters mostly from different settlements.

where $|S|$ is the number of nodes in the settlement $S$ and $\{alters_i\}$ is the subgraph of the alters of node $i \in S$. This measure captures the typical diversity of social perspectives that the members of the settlement access. At the settlement level, this measure captures bridging social capital.

Settlement diversity is positively correlated with share of the population graduating from high school ($\rho \approx 0.62$) and average income ($\rho \approx 0.55$). Fragmentation and diversity are positively correlated ($\rho \approx 0.46$), which is not surprising given that both are calculated using network modularity. However, the ego-focus and, more importantly, the inclusion of inter-settlement edges of the diversity measure distinguish it from fragmentation (see the electronic supplementary material). Despite this correlation, we observe that they predict different corruption outcomes.

## 2.3. Models

The primary aim of our paper is to relate bonding and bridging social capital in settlements to corruption risk in their public contracts. Our hypothesis H1 is related to excess bonding social capital, measured by fragmentation, while hypothesis H2 refers to surplus in bridging social capital, measured by diversity. We predict average contract corruption risk at the settlement level using ordinary least squares (OLS) multiple regressions of the following form:

$$C_S = \beta_1 * F_S + \beta_2 * D_S + X_S * \theta + \epsilon_S,$$

where $C_S$ is one of two corruption risk indicators, averaged at the settlement level, $F_S$ is the settlement's fragmentation, $D_S$ is the settlement's diversity, $X_S$ is a matrix of control variables defined below and $\epsilon_S$ is an error term. The $\beta$s are scalar and $\theta$ a vector of unknown parameters.

We include a variety of control variables in our regressions. Past research has found significant relationships between wealth, education, employment and corruption [2], so we control for settlement average income, its share of high school graduates, the presence of a university in the settlement, and its unemployment and inactivity rates. As demographic features of settlements may influence the measured social network, we include total population, rate of iWiW use and share of the population over 60 in our models [61]. These socio-economic controls indicate 2011 levels when possible, as 2011 was the most recent Hungarian census. We also control for the settlement's mayor's average victory margin in the 2002, 2006 and 2010 elections as a proxy for the level of political competition in the settlement, which has been found to be positively related with local quality of government [62]. Finally, we include a geographical feature of the settlements: the minimum travel distance in minutes from the capital, Budapest. Past work indicates that distance from central authorities predicts higher rates of corruption [63]. We present additional details on the control variables in the electronic supplementary material. For the sake of comparison, we fit a baseline model including only the control terms.

Implicit in our modelling framework is our choice to aggregate the social network measures, corruption risk scores and controls of settlements into a single snapshot. Contracts range from 2006 to 2014, iWiW friendships from 2002 to 2012, and controls are set at 2011 levels (corresponding to the last Hungarian census). As we are studying the relationship between social structure and corruption, both long-run phenomena, we claim that this represents sufficient temporal overlap.

## 3. Results

We summarize our findings in table 2. We see that there is a significant relationship between social network structure and both dependent variables measuring corruption. More fragmentation consistently predicts more corruption, while more diversity consistently predicts less corruption. In both cases, adding the network features significantly improves the adjusted $R^2$ of the model. Moreover, comparing the coefficients, we see that the social network features have effect sizes comparable to that of any social, political or economic control. We present alternative specifications and robustness checks in the electronic supplementary material, including the intermediate models containing only one network feature. All models pass a variance inflation factor (VIF) test for feature collinearity (see the electronic supplementary material).

We visualize the effects of our network variables in figure 5. We plot model-predicted rate of closed procedure or single-bid contract awards ($C_{csb}$) including 90% confidence intervals for varying levels of fragmentation and average ego diversity. As the variables are standardized, the units can be interpreted as standard deviations from the mean (at 0). We observe that, all else equal, our model predicts that going from one standard deviation below average fragmentation to one standard deviation above average, increases $C_{csb}$ by about one half of a standard deviation. Diversity has a stronger effect in the other direction: the same change (from one standard deviation below average to one above average) induces a full standard deviation decrease in the corruption indicator. The effect

**Table 2.** Settlement-level regression results predicting two corruption risk indicators. For both dependent variables, the first columns (1) and (3) correspond to the base model, predicting corruption risk using only control variables, and the second columns (2) and (4) show results, when the social network features are included. Note that all features are standardized with mean 0 and standard deviation 1.

| dependent variable: | % closed or single bid | | average CRI | |
|---|---|---|---|---|
| | (1) | (2) | (3) | (4) |
| *fragmentation* | | 0.263*** | | 0.207** |
| (bonding social capital) | | (0.097) | | (0.092) |
| *diversity* | | −0.553*** | | −0.551*** |
| (bridging social capital) | | (0.176) | | (0.168) |
| income/capita | −0.262 | −0.277* | −0.075 | −0.096 |
| | (0.169) | (0.162) | (0.161) | (0.155) |
| N contracts (log) | −0.313* | −0.314* | −0.685*** | −0.697*** |
| | (0.171) | (0.165) | (0.162) | (0.158) |
| population (log) | −0.180 | 0.020 | 0.118 | 0.335** |
| | (0.143) | (0.166) | (0.136) | (0.159) |
| rate iWiW use | 0.045 | 0.037 | 0.122 | 0.107 |
| | (0.137) | (0.132) | (0.130) | (0.126) |
| mayor victory margin | 0.278*** | 0.255*** | 0.303*** | 0.281*** |
| | (0.089) | (0.086) | (0.085) | (0.082) |
| % high school grads | 0.166 | 0.374* | −0.176 | 0.040 |
| | (0.190) | (0.199) | (0.181) | (0.190) |
| distance to Budapest | −0.021 | −0.198* | 0.061 | −0.112 |
| | (0.104) | (0.112) | (0.099) | (0.107) |
| share of pop. inactive | −0.797*** | −0.805*** | −0.716*** | −0.754*** |
| | (0.229) | (0.229) | (0.218) | (0.219) |
| unemployment rate | 0.239** | 0.262** | 0.299*** | 0.320*** |
| | (0.118) | (0.113) | (0.112) | (0.108) |
| % population 60+ | 0.501*** | 0.491*** | 0.500*** | 0.503*** |
| | (0.163) | (0.158) | (0.155) | (0.151) |
| has university | 0.351 | 0.294 | 0.431** | 0.352* |
| | (0.220) | (0.221) | (0.210) | (0.211) |
| constant | 1.245* | 1.206* | 2.779*** | 2.790*** |
| | (0.725) | (0.702) | (0.689) | (0.671) |
| observations | 169 | 169 | 169 | 169 |
| adjusted $R^2$ | 0.163 | 0.230 | 0.183 | 0.243 |
| F statistic | 3.967*** | 4.859*** | 4.419*** | 5.142*** |

Significance thresholds: *$p < 0.1$; **$p < 0.05$; ***$p < 0.01$.

of the network features on $C_{CRI}$ is similar. In the electronic supplementary material, we present an ANOVA feature importance test that indicates the significance of both network-based features.

## 4. Discussion

In this paper, we used data from an online social network and a collection of public procurement contracts to relate the social capital of Hungarian settlements to the corruption in its local

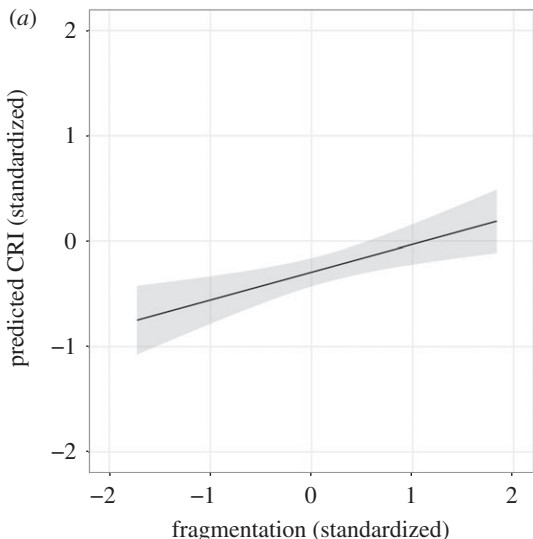
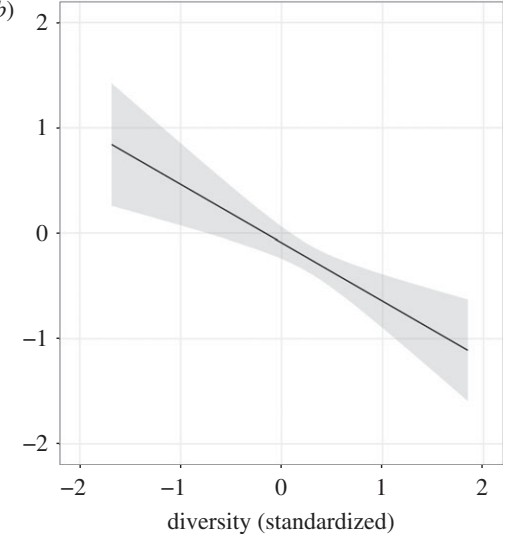

**Figure 5.** Plots of marginal effects of the key social capital variables and their predicted impact on a settlement's rate of closed procedure or single bidder contract awards; shaded regions represent 90% confidence intervals. As the variables are standardized, unit changes on either axis can be interpreted as standard deviation changes. Fragmentation (*a*), quantifying excess bonding social capital in a community, predicts higher corruption risk, while diversity (*b*) predicts lower corruption risk.

government. To our knowledge, this paper is the first to study social aspects of corruption using large-scale social network data.

We introduced measures to quantify excess bonding and bridging social capital at the settlement level from online social network data. We found that settlements with high bonding social capital tend to award contracts with higher corruption risk. We also found that settlements with high bridging social capital tend to award lower corruption risk contracts. Social capital measures add substantive predictive power to models of corruption outcomes, above baseline models controlling for other socio-economic factors such as average income, education, political competition and demography.

We recognize several limitations to our approach. An inherent challenge in the research of corruption is that proven cases are rare, and so our measures can only track risk or suspicion of corruption. Moreover, we assume that steering contracts to certain firms by bureaucrats indicates corruption—but it may happen that bureaucrats make socially optimal decisions using their local knowledge of markets and discretion [64].

It is also clearly the case that iWiW is not a full map of social relations in Hungary and its users do not make up a representative sample of the population. Finally, we do not claim to have found a causal link between social capital and corruption risk. Besides the potential of omitted variable bias, it is highly likely that corruption also influences social capital in the long run [65].

Despite these limitations, we believe that our findings are valuable. Above all, our novel, data-based, settlement-level approach provides new evidence for the old hypothesis that corruption is a structural phenomenon. Our finding that social structure relates to corruption risk suggests, for example, why appointing an ombudsman in a corrupt place rarely improves corruption outcomes [2] and why anti-corruption laws can backfire if they conflict with prevailing social norms [66].

That is not to say that fighting corruption is futile. Rather we believe our findings suggest that top-down efforts are unlikely to work unless they impact social capital or other significant covariates of our model like political competition. Our conclusions hint at potential mechanisms which sustain corruption. Factors, such as racial segregation or economic inequality, which may drive fragmentation are ideal targets for policy interventions [6].

Data accessibility. As the iWiW data are protected under an NDA, we are only able to share settlement-level aggregated data of network features. Aggregated data and code to replicate our models and results are deposited at the Dryad Digital Repository at: https://doi.org/10.5061/dryad.jb48dg0 [67].

Authors' contributions. J.W. and J.K. conceived of the presented idea. J.W. and B.L. collected data on municipalities. J.W. and T.Y. developed the methods used. J.W. analysed the data. All authors contributed to writing the manuscript and gave final approval for publication.

Competing interests. The authors declare no competing interests.

Funding. T.Y. was partially supported by the Alan Turing Institute under the EPSRC grant no. EP/N510129/1. B.L. and J.K. gratefully acknowledge the financial support from the Hungarian Scientific Research Fund (OTKA K129124, 'Uncovering patterns of social inequalities and imbalances in large-scale networks').

Acknowledgements. The authors thank Bernie Hogan, Ralph Schroeder, David Deritei, Mihály Fazekas, Marc Sarazin, Cohen Simpson, Bharath Ganesh, participants of seminars at Oxford and the Hungarian Academy of Sciences, and anonymous referees for their valuable comments. The authors are grateful to János Török for assistance with the iWiW data, and to Ágnes Czibik and Mihály Fazekas for the assistance with the public contracting data.

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
