## [Reviewer comments · Royal Society Open Science]

Review History

RSOS-182103.R0 (Original submission)

Review form: Reviewer 1

Is the manuscript scientifically sound in its present form?

Yes

Are the interpretations and conclusions justified by the results?

Yes

Is the language acceptable?

Yes

Is it clear how to access all supporting data?

Yes

Do you have any ethical concerns with this paper?

No

Have you any concerns about statistical analyses in this paper?

No

Recommendation?

Accept as is

Comments to the Author(s)

I am satisfied with how authors addressed my concerns

Review form: Reviewer 2

Is the manuscript scientifically sound in its present form?

Yes

Are the interpretations and conclusions justified by the results?

Yes

Is the language acceptable?

Yes

Is it clear how to access all supporting data?

Yes

Do you have any ethical concerns with this paper?

No

Have you any concerns about statistical analyses in this paper?

No

Recommendation?

Accept as is

Comments to the Author(s)

This manuscript investigates the relationship between social capital and corruption risk at the level of a settlement, making a contribution to the larger corruption literature. It was in pretty good shape when I first read it for Interface; it is in excellent shape now. The authors assiduously addressed my comments, from the smallest recommendations of a particular citation to the largest requests for clarification and robusticity checks. I especially appreciated their inclusion of Supplementary Figure 6, visually illustrating the logic between their cut-off point for iWiW users with many connections, and their re-runs of the models (1) excluding one term that led to a borderline VIF and (2) excluding users above different numbers of connections inspired by Dunbar's work. I made the smallest of punctuation and wording recommendations in the attached file (Appendix A). With those punctuation and wording changes made, I recommend this piece for publication.

Decision letter (RSOS-182103.R0)

01-Mar-2019

Dear Dr Wachs

On behalf of the Editors, I am pleased to inform you that your Manuscript RSOS-182103 entitled "Social capital predicts corruption risk in towns" has been accepted for publication in Royal Society Open Science subject to minor revision in accordance with the referee suggestions. Please find the referees' comments at the end of this email.

The reviewers and handling editors have recommended publication, but also suggest some minor revisions to your manuscript. Therefore, I invite you to respond to the comments and revise your manuscript.

- Ethics statement

- Data accessibility

<http://datadryad.org/submit?journalID=RSOS&manu=RSOS-182103>

- Competing interests

- Authors' contributions

AB carried out the molecular lab work, participated in data analysis, carried out sequence alignments, participated in the design of the study and drafted the manuscript; CD carried out

the statistical analyses; EF collected field data; GH conceived of the study, designed the study, coordinated the study and helped draft the manuscript. All authors gave final approval for publication.

- Acknowledgements

- Funding statement

Because the schedule for publication is very tight, it is a condition of publication that you submit the revised version of your manuscript before 10-Mar-2019. Please note that the revision deadline will expire at 00.00am on this date. If you do not think you will be able to meet this date please let me know immediately.

- 1) A text file of the manuscript (tex, txt, rtf, docx or doc), references, tables (including captions) and figure captions. Do not upload a PDF as your "Main Document";
- 2) A separate electronic file of each figure (EPS or print-quality PDF preferred (either format should be produced directly from original creation package), or original software format);
- 3) Included a 100 word media summary of your paper when requested at submission. Please ensure you have entered correct contact details (email, institution and telephone) in your user account;
- 4) Included the raw data to support the claims made in your paper. You can either include your data as electronic supplementary material or upload to a repository and include the relevant doi within your manuscript. Make sure it is clear in your data accessibility statement how the data can be accessed;

5) All supplementary materials accompanying an accepted article will be treated as in their final form. Note that the Royal Society will neither edit nor typeset supplementary material and it will be hosted as provided. Please ensure that the supplementary material includes the paper details where possible (authors, article title, journal name).

on behalf of Prof Miles Padgett (Subject Editor)
openscience@royalsociety.org

Associate Editor Comments to Author:

The reviewers are largely satisfied with your manuscript, but recommend a few minor tweaks -- otherwise, good job! Well done!

Reviewer comments to Author:

Reviewer: 1

Comments to the Author(s)

I am satisfied with how authors addressed my concerns

Reviewer: 2

Comments to the Author(s)

This manuscript investigates the relationship between social capital and corruption risk at the level of a settlement, making a contribution to the larger corruption literature. It was in pretty

good shape when I first read it for Interface; it is in excellent shape now. The authors assiduously addressed my comments, from the smallest recommendations of a particular citation to the largest requests for clarification and robusticity checks. I especially appreciated their inclusion of Supplementary Figure 6, visually illustrating the logic between their cut-off point for iWiW users with many connections, and their re-runs of the models (1) excluding one term that led to a borderline VIF and (2) excluding users above different numbers of connections inspired by Dunbar's work. I made the smallest of punctuation and wording recommendations in the attached file. With those punctuation and wording changes made, I recommend this piece for publication.

Author's Response to Decision Letter for (RSOS-182103.R0)

See Appendix B.

Decision letter (RSOS-182103.R1)

12-Mar-2019

Dear Dr Wachs,

I am pleased to inform you that your manuscript entitled "Social capital predicts corruption risk in towns" is now accepted for publication in Royal Society Open Science.

on behalf of Professor Miles Padgett (Subject Editor)
openscience@royalsociety.org

Appendix A**Social capital predicts corruption risk in towns**

Journal:	Royal Society Open Science
Manuscript ID	RSOS-182103
Article Type:	Research
Date Submitted by the Author:	14-Dec-2018
Complete List of Authors:	Wachs, Johannes; Central European University, Department of Network and Data Science Yasseri, Taha; University of Oxford, Oxford Internet Institute Lengyel, Balazs; Hungarian Academy of Sciences Kertész, János; Central European University, Department of Network and Data Science; Budapest University of Technology and Economics Institute of Physics
Subject:	e-science < COMPUTER SCIENCE, Complexity < PHYSICS
Keywords:	Social Networks, Corruption, Social Capital
Subject Category:	Physics

Dear Editor,

We write to submit our article *Social capital predicts corruption risk in towns* for publication in **Royal Society: Open Science**. We were invited to transfer a revised version of our manuscript originally submitted to Royal Society: Interface (id: rsif-2018-0762). We first wish to acknowledge the editor and reviewers of our paper for their helpful comments and the invitation to transfer the submission to your journal. The updated version of our paper addresses all points made by the two reviewers and the editor concerning our earlier submission.

The attached memo describes these changes in detail. For the sake of readability, we assign progressive code numbers to comments. We discuss the comment and suggest how we address the issue in the paper, quoting directly from the revised text when appropriate.

The most significant changes are:

1. We give a careful rewording of our claims and discussion to avoid misinterpretation related to causality in our findings.
2. Thanks to references suggested by Reviewer 1, we have improved our review of the literature and made modifications to our framing.
3. We have improved clarity, especially we explain why we use a snapshot of the social network data and why we aggregate our contract corruption risk scores.
4. We carried out robustness checks against thresholds we used to filter our data, finding that our results remain intact. We note these alternative specifications in the text and include additional tables and figures in the SI.

We believe that our manuscript has considerably improved and hope that it is now publishable in your journal. We look forward to hearing from you in due course.

Kind regards,

Johannes Wachs, Taha Yasseri, Balazs Lengyel, and Janos Kertesz

Referee: 1

Ref1Comment1: *In the attached file I have included comments and suggested edits throughout the text and supplement. Many of these are minor points, requesting clarification for the reader or suggesting alternate word choice.*

We wish to thank Reviewer 1 for her/his positive opinion about our paper and the detailed suggestions for improvement of the text. We have addressed all minor points in the marked-up copy of our manuscript. Among several useful references, we are especially grateful for calling our attention to the text by Yamagichi, which led us to a significant improvement in our theoretical framing. Though we addressed comments through, the most significant changes are in the introduction.

R1C2:- *The choices of exclusion criteria are not sufficiently motivated. Excluding towns with fewer than 5 contracts issued during the study period is likely to bias against small towns; per the authors' point in the Discussion that towns with fewer contracts may be especially likely to fill contracts in a corrupt manner, given lack of a smooth-running system for assessment, the inclusion of these towns should only amplify their effect. Likewise, the exclusion of participants with more than 10,000 connections should be better motivated, as the cut-off seems arbitrary and may bias against users from big towns. In short, I worry that both ends of the distribution have been truncated due to these decisions, so more motivation is needed to assure the reader that these decisions are justified -- or these data should be included in the dataset and the results re-run. Clearer justifications are also needed for the exclusion of Budapest and the use of average income (as opposed to Gini or median income).*

We acknowledge that in the original manuscript the chosen cutoffs were poorly motivated and the choice of control variables deserved more detail and thank the reviewer for pointing these out. We address the points in order:

1. We removed towns with fewer than five contracts per year for three reasons.
 - a. The first is that our corruption risk indicators are noisy measures of potential bad behavior. When aggregated over many contracts, the signal becomes clear (i.e. 80% closed procedure or single bidding vs 20%).
 - b. The second is that in addition to the stylized fact that corruption may be easier in small towns (given the lower stakes, lower quality of oversight), other factors may impact our red flag indicators in this setting. Smaller settlements may be served by fewer firms (which may increase the likelihood of single bidding). Towns issuing few contracts have less practice in the procedures and may make mistakes (i.e. waiting until the last moment, requiring a short notice period and attracting a single bidder).

To test the robustness of our findings to this threshold, we rerun our analysis twice, including all settlements with at least 1, respectively at least 2, contracts per year. We find that the main findings (fragmentation is positively and diversity negatively related to settlement level corruption risk

indicators) of our models are preserved when including these settlements,
we report these model tables in the appendix.

2. We removed users with over 10,000 connections because we are interested in quantifying social connections between individuals. **Reviewer 2** would have preferred a significantly lower threshold, suggesting that it is unlikely that any individual can have more than 10,000 genuine connections with other people. As we argue in the revised text (in subsection: Measuring social capital): *this cutoff balances two concerns: it excludes those accounts with so many connections that it brings into question the nature of its connections, and we avoid truncating the tail of the distribution of social connectivity too much, allowing for sociality to range over several orders of magnitude.* We also cite a paper on detecting fake accounts on social media which finds that high node degree is a particularly useful filter. We also note that we take the same approach as Lengyel et al (2015). We have included a short discussion about the filtering to make our position clear. We added a test of our choice of threshold to the appendix, finding that both fragmentation and diversity of towns are not significantly changed (i.e. within 5% of the measures used in the paper) when excluding users with more than 500, 1,000, 2,000, or 10,000 connections.
 3. We improved our explanation for why we excluded Budapest, noting the difference in political organization. Quoting from the revision (in subsection Local Government Contracting Data): *We exclude Budapest for two reasons: it is a severe outlier in size and economic importance and because of its unique governance structure. Budapest is split into 23 districts, each with its own local government and mayor. It also has a city-wide government and mayor. As iWiW does not distinguish between districts and that many contracting decisions are taken at the district level, we judged that we could not reasonably compare the full city with other settlements in Hungary.*

R1C3: *It is unclear why the authors chose to average across years both for contracts and for iWiW data. In the Discussion, they mention that they cannot discern whether corruption drives social network structure or vice versa with these data, but it seems to me that they can. Since data are available for both iWiW and contracts across about a decade, the authors could analyze the contract values by year for each town, including a random effect for town, and assess whether lag -1 or lag +1 values for social structure predict the contract data. This may not be possible, however, due to the coverage of their data -- and this coverage needs to be made more clear in the manuscript, regardless of whether the authors choose to change their models or not. It was unclear whether iWiW data came from all years or not, and what year control data come from (except for margin of victory in mayoral elections, which is clearly stated). I can imagine some cause for concern, for example, if unemployment rate data come from 2013 but the majority of contracts represented in a town's average were awarded in 2007. The authors need to convince the readers that the same years are covered, and if they are not, that this non-overlapping coverage is not cause for concern.*

These points were very helpful and we hope that resolving them improved the clarity of the paper. Our primary limitation in this regard is the iWiW data and particularly two of its

properties: first that the site was growing almost continuously until the start of its demise in
2012, with new users joining; second that links do not encode activity (unlike connections in
mobile phone data). Because new users are joining, it is difficult to tell if a settlement's
network is changing qualitatively (becoming more diverse or fragmented) when new
connections are made, or if it is just an improvement in the accuracy of the picture because
an old friendship is now represented in the data.

More generally, we are trying to make a claim about a long-run phenomenon, motivated by
the observation that corruption risk is clustered across time and in space. Even if a town
becomes more diverse or fragmented, we would expect a significant lag before the quality
of governance was improved. Finally, we made sure to document the precise years for
which we have control data.

**R1C4:** *Some clarity is needed with terminology, including identifying which words are used*
*interchangeably (e.g., community and group are the same, and settlement and town are the*
*same...?) and words that are discipline-specific (e.g., what is a bubble?).*

This is again a useful comment. We have edited the manuscript to use only settlement (no
town, municipality, city, etc) and community (only for algorithmically detected groups of
nodes within the iWiW network). We have included a sentence clarifying this point in the
text. We have also revised the sentence about the asset bubble.

**R1C5:** *More clarity is needed in the variable descriptions in Table 1 and the SI. Some of the*
*criteria for variables (e.g., above average word length of calls for contract bids) are unclear*
*and seem potentially ad hoc without more justification.*

In the interest of space and flow of the manuscript we have supplemented our discussion of
the features in the supplementary materials. Nevertheless, we do acknowledge that we are
significantly leaning on past work using these measures, for instance considering above
average participation requirement text length as a red flag (rather than 1 or 2 standard
deviations above average) is indeed a choice, following the approach of Fazekas and Kocsis
(2017), who find a significant signal using this threshold.

Referee: 2

**Ref2Comment1:** *The evidences provided do not support the proposed hypotheses H1 and*
*H2. In fact, it is quite possible, and plausible, that the presence of corruption lead to more*
*fragmentation and less diversity (tribalism). At best, the evidences provided establish*
*correlation. Authors very briefly point out to this limitation in the discussion, as they conquer*
*that no causal link was established by the provided evidence. Yet, the hypothesis H1 implies*
*causal link, through the word "enables". The same is true for H2 through the word*
*"suppresses". While this does not null the value, not the importance, of the work and effort*
*provided in this paper, I think it is important to reword the hypotheses to avoid any causal*
*claim (unless authors can find some instrumental variable that help establish the causality).*

We are grateful to the referee for pointing this out and we acknowledge that in our original
draft our claims seem to imply a causal relationship. We have significantly rewritten the
relevant parts of the text to avoid this implication.

**R2C2:** *In the supplementary material, authors explain that they omit users with more than*
*10,000 connection, since these are hardly social ties. Why particularly 10,000? I can*
*confidently say that no one would remember the names of 1000 people at the same time!*
*This part needs at least some evidence of the literature.*

First we would like to acknowledge that the reviewer makes a good point: no one can have
10,000 friends, let alone 1,000 (consider Dunbar). On the other hand we note that Reviewer
1 suggested that we are cutting off interesting users in the long tail which may have an
interesting effect on the structural estimates. Such users may also provide a way for friends
to meet online (for example if both of us are friends with such a super user, we may be
more likely to encounter each other's profiles). Please consider our response to Reviewer 1
(R1C2) and the following addition to the text: *this cutoff balances two concerns: it excludes*
*those accounts with so many connections that it brings into question the nature of its*
*connections, and we avoid truncating the tail of the distribution of social connectivity too*
*much, allowing for sociality to range over several orders of magnitude.*

We also carried out a robustness test, comparing the fragmentation and diversity of
settlements as the cutoff ranges across the values: 100, 250, 500, 1000, 2000, 10000. We
find that the measures in the paper (i.e. with a cutoff at 10,000) are within 5% of alternative
choices of the threshold at or above 500 connections. We report these findings in the
supplementary information.

**R2C3:** *How can readers access the data to verify the analysis done in the paper? This is*
*important, but was not clear. A related point is the privacy of the data. When describing the*
*use of social network data, authors (correctly) raised the issue of user privacy. However, no*
*details were provided in the supplementary material (how was the data protected and*
*privacy insured). This is particularly important if the data is to be made public.*

We thank Reviewer 2 for reminding us of the importance of this aspect of our work. We
obtained access to the social network data through agreement with the owner on condition
of signing an NDA. We received the data with names removed and replaced with
anonymous IDs. Moreover, the terms of the NDA do not allow us to share individual level
data, hence we share only town-level aggregate features from the social network
(fragmentation, diversity, user rate). We have update the data availability statement to be
more clear, and note that we included settlement-level data in our original submission.

Social capital predicts corruption risk in towns

Johannes Wachs^a, Taha Yasseri^{b,c}, Balázs Lengyel^{d,e}, and János Kertész^{a,f}

[revised manuscript text omitted]

References

- [1] D. Acemoglu and M. O. Jackson. Social norms and the enforcement of laws. *Journal of the European Economic Association*, 15(2):245–295, 2017.

[2] A. Alesina and E. Zhuravskaya. Segregation and the quality of government in a cross section of
countries. *American Economic Review*, 101(5):1872–1911, 2011.
- [3] K. Attström, R. Krber, and M. Junclaus. Review of the Function of the CPV Codes/System.
Technical report, European Commission, 2012.
- [4] M. Bailey, R. Cao, T. Kuchler, J. Stroebel, A. Wong, et al. Social connectedness: Measurement,
determinants, and effects. *Journal of Economic Perspectives*, 32(3):259–80, 2018.
- [5] C. Bjørnskov. Corruption and social capital. Technical report, University of Aarhus, Aarhus
School of Business, Department of Economics, 2003.
- [6] V. D. Blondel, J.-L. Guillaume, R. Lambiotte, and E. Lefebvre. Fast unfolding of communities in
large networks. *Journal of statistical mechanics: theory and experiment*, 2008(10):P10008, 2008.
- [7] E. Bokányi, D. Kondor, L. Dobos, T. Sebők, J. Stéger, I. Csabai, and G. Vattay. Race, religion
and the city: twitter word frequency patterns reveal dominant demographic dimensions in the
united states. *Palgrave Communications*, 2:16010, 2016.
- [8] R. Broms, C. Dahlström, and M. Fazekas. Procurement and Competition in Swedish Municipal-
ities. *QOG Working Paper Series*, (5), 2017.
- [9] B. Brooks, B. Hogan, N. Ellison, C. Lampe, and J. Vitak. Assessing structural correlates to social
capital in facebook ego networks. *Social Networks*, 38:1–15, 2014.
- [10] Campante, Filipe R and Do, Quoc-Anh. Isolated capital cities, accountability, and corruption:
Evidence from US states. *American Economic Review*, 104(8):2456–81, 2014.
- [11] Q. Cao, M. Sirivianos, X. Yang, and T. Pregueiro. Aiding the detection of fake accounts in large
scale social online services. In *Proceedings of the 9th USENIX conference on Networked Systems
Design and Implementation*, pages 15–15. USENIX Association, 2012.
- [12] N. Charron, L. Dijkstra, and V. Lapuente. Regional governance matters: Quality of government
within european union member states. *Regional Studies*, 48(1):68–90, 2014.
- [13] N. Charron, C. Dahlström, M. Fazekas, and V. Lapuente. Careers, connections, and corruption
risks: Investigating the impact of bureaucratic meritocracy on public procurement processes. *The
Journal of Politics*, 79(1):89–104, 2017.
- [14] J. S. Coleman. Social capital in the creation of human capital. *American Journal of Sociology*,
pages S95–S120, 1988.
- [15] D. Coviello, A. Guglielmo, and G. Spagnolo. The effect of discretion on procurement performance.
*Management Science*, 64(2):715–738, 2017.
- [16] R. I. Dunbar, V. Arnaboldi, M. Conti, and A. Passarella. The structure of online social networks
mirrors those in the offline world. *Social Networks*, 43:39–47, 2015.
- [17] N. Eagle, M. Macy, and R. Claxton. Network diversity and economic development. *Science*, 328
(5981):1029–1031, 2010.
- [18] B. H. Erickson. Secret societies and social structure. *Social Forces*, 60(1):188–210, 1981.
- [19] European Commission. The OLAF Report 2017, p. 15. 2018. Available at URL:
https://ec.europa.eu/anti-fraud/sites/antifraud/files/olaf_report_2017_en.pdf.
- [20] M. Fazekas and G. Kocsis. Uncovering high-level corruption: Cross-national objective corruption
risk indicators using public procurement data. *British Journal of Political Science*, pages 1–10,
2017.

[21] M. Fazekas and B. Tóth. The extent and cost of corruption in transport infrastructure. new
evidence from europe. *Transportation Research Part A: Policy and Practice*, 113:35–54, 2018.
[22] M. Fazekas, I. J. Tóth, and L. P. King. An objective corruption risk index using public procure-
ment data. *European Journal on Criminal Policy and Research*, 22(3):369–397, 2016.
[23] M. Fazekas, J. Skuhrovec, and J. Wachs. Corruption, government turnover, and public contracting
market structure—insights using network analysis and objective corruption proxies. *GTI Working
Paper Series*, (2), 2017.
[24] M. Fazekas, R. Ferrali, and J. Wachs. Institutional quality, campaign contributions, and
favouritism in us federal government contracting. *GTI Working Paper Series*, (1), 2018.
[25] J. D. Fearon and D. D. Laitin. Explaining interethnic cooperation. *American political science
review*, 90(4):715–735, 1996.
[26] D. Gambetta. *The Sicilian Mafia: the Business of Private Protection*. Harvard University Press,
1996.
[27] C. Geertz. *Peddlers and princes: Social development and economic change in two Indonesian
towns*, volume 318. University of Chicago Press, 1963.
[28] E. L. Glaeser and R. E. Saks. Corruption in America. *Journal of Public Economics*, 90(6-7):
1053–1072, 2006.
[29] M. S. Granovetter. The strength of weak ties. *American Journal of Sociology*, 78(6):1360–1380,
1973.
[30] L. Guiso, P. Sapienza, and L. Zingales. Civic capital as the missing link. In *Handbook of Social
Economics*, volume 1, pages 417–480. Elsevier, 2011.
[31] J. F. Hair, R. E. Anderson, R. L. Tatham, W. C. Black, et al. *Multivariate Analysis*. Prentice
Hall, 1999.
[32] D. Harris. Bonding social capital and corruption: A cross-national empirical analysis. Technical
report, University of Cambridge, Department of Land Economics, 2007.
[33] T. International. *Global Corruption Report 2007*. Cambridge University Press, 2007.
[34] S. Knack. Social capital and the quality of government: Evidence from the states. *American
Journal of Political Science*, pages 772–785, 2002.
[35] B. Lancee. The Economic Returns of Immigrants Bonding and Bridging Social Capital: The
Case of the Netherlands. *International Migration Review*, 44(1):202–226, 2010.
[36] J. Laurence. The effect of ethnic diversity and community disadvantage on social cohesion: A
multi-level analysis of social capital and interethnic relations in UK communities. *European
Sociological Review*, 27(1):70–89, 2009.
[37] B. Lengyel, A. Varga, B. Ságvári, Á. Jakobi, and J. Kertész. Geographies of an online social
network. *PLoS one*, 10(9):e0137248, 2015.
[38] S. S. Levine, E. P. Apfelbaum, M. Bernard, V. L. Bartelt, E. J. Zajac, and D. Stark. Ethnic
diversity deflates price bubbles. *Proceedings of the National Academy of Sciences*, 111(52):18524–
18529, 2014.
[39] N. Lin. *Social capital: A theory of social structure and action*, volume 19. Cambridge University
Press, 2002.

[40] L. Lorincz, J. Koltai, A. F. Gyor, K. Takacs, et al. Collapse of an online social network: The blame
on social capital. Technical report, Institute of Economics, Centre for Economic and Regional
Studies, Hungarian Academy of Sciences, 2017.
- [41] M. W. Macy and J. Skvoretz. The evolution of trust and cooperation between strangers: A
computational model. *American Sociological Review*, pages 638–660, 1998.
- [42] M. Mamei, F. Pancotto, M. De Nadai, B. Lepri, M. Vescovi, F. Zambonelli, and A. Pentland. Is
social capital associated with synchronization in human communication? an analysis of italian
call records and measures of civic engagement. *EPJ Data Science*, 7(1):25, 2018.
- [43] P. Mauro. The persistence of corruption and slow economic growth. *IMF staff papers*, 51(1):
1–18, 2004.
- [44] A. Mungiu-Pippidi. Controlling corruption through collective action. *Journal of Democracy*, 24
(1):101–115, 2013.
- [45] M. E. Newman. Modularity and community structure in networks. *Proceedings of the National
Academy of Sciences*, 103(23):8577–8582, 2006.
- [46] L. Norbutas and R. Corten. Network structure and economic prosperity in municipalities: A
large-scale test of social capital theory using social media data. *Social Networks*, 52:120–134,
2018.
- [47] L. Norman and A. Komuves. EU Fraud Office Finds Irregularities in Projects
Linked to Hungarian Leaders Son-in-Law. *The Wall Street Journal*, Jan 2018. URL
[https://www.wsj.com/articles/eu-fraud-office-finds-irregularities-in-projects-
linked-to-hungarian-leaders-son-in-law-1515744340](https://www.wsj.com/articles/eu-fraud-office-finds-irregularities-in-projects-linked-to-hungarian-leaders-son-in-law-1515744340).
- [48] OECD.Stat. Government at a glance - 2017 edition: Public procurement. [https://
stats.oecd.org/Index.aspx?QueryId=78413](https://stats.oecd.org/Index.aspx?QueryId=78413), 2017. Accessed: 2018-08-09.
- [49] B. A. Olken. Corruption perceptions vs. corruption reality. *Journal of Public economics*, 93(7-8):
950–964, 2009.
- [50] M. Paccagnella and P. Sestito. School cheating and social capital. *Education Economics*, 22(4):
367–388, 2014.
- [51] A. Persson, B. Rothstein, and J. Teorell. Why anticorruption reforms failsystemic corruption as
a collective action problem. *Governance*, 26(3):449–471, 2013.
- [52] U. Pfeil, R. Arjan, and P. Zaphiris. Age differences in online social networking—a study of user
profiles and the social capital divide among teenagers and older users in myspace. *Computers in
Human Behavior*, 25(3):643–654, 2009.
- [53] A. Portes. Social capital: Its origins and applications in modern sociology. *Annual Review of
Sociology*, 24(1):1–24, 1998.
- [54] A. Portes. The two meanings of social capital. *Sociological Forum*, 15:1–12, 2000.
- [55] A. Portes. Downsides of social capital. *Proceedings of the National Academy of Sciences*, 111
(52):18407–18408, 2014.
- [56] R. D. Putnam. *Bowling alone: The collapse and revival of American community*. Simon and
Schuster, 2001.
- [57] PwC EU Services. Identifying and Reducing Corruption in Public Procurement in the EU.
Technical report, European Commission, 2013.

[58] S. Richey. The impact of corruption on social trust. *American Politics Research*, 38(4):676–690,
2010.
[59] B. Rothstein and E. M. Uslaner. All for all: Equality, corruption, and social trust. *World Politics*,
58(1):41–72, 2005.
[60] P. Sah, S. T. Leu, P. C. Cross, P. J. Hudson, and S. Bansal. Unraveling the disease consequences
and mechanisms of modular structure in animal social networks. *Proceedings of the National*
*Academy of Sciences*, page 201613616, 2017.
[61] F. Santo and B. Marc. Resolution limit in community detection. *Proceedings of the National*
*Academy of Sciences*, 104:36–41, 2007.
[62] H.-E. Sung. Democracy and political corruption: A cross-national comparison. *Crime, Law and*
*Social Change*, 41(2):179–193, 2004.
[63] J. Török and J. Kertész. Cascading collapse of online social networks. *Scientific Reports*, 7(1):
16743, 2017.
[64] J. Török, Y. Murase, H.-H. Jo, J. Kertész, and K. Kaski. What big data tells: sampling the
social network by communication channels. *Physical Review E*, 94(5):052319, 2016.
[65] D. Torsello and B. Venard. The anthropology of corruption. *Journal of Management Inquiry*, 25
(1):34–54, 2016.
[66] J. Wachs, T. Yasseri, B. Lengyel, and J. Kertész. Data from: Social capital predicts corruption
risk in towns, 2018.
[67] M. Weber. *Economy and society: An outline of interpretive sociology*, volume 1. Univ of California
Press, 1978.
[68] O. Weisel and S. Shalvi. The collaborative roots of corruption. *Proceedings of the National*
*Academy of Sciences*, 112(34):10651–10656, 2015.
[69] H. Westlund and F. Adam. Social capital and economic performance: A meta-analysis of 65
studies. *European Planning Studies*, 18(6):893–919, 2010.
[70] D. Williams. On and off thenet: Scales for social capital in an online era. *Journal of Computer-*
*Mediated Communication*, 11(2):593–628, 2006.
[71] T. Yamagishi. *Trust: The evolutionary game of mind and society*. Springer, 2011.

1 Supplementary Information

1.1 Description of iWiW data

In line with previous work on iWiW we filtered the data used in our analysis. We use the data from the network at its peak activity in 2012. Out of roughly 4.5 million user accounts, we dropped the roughly 500,000 accounts with location outside of Hungary. We follow Lengyel et al. [37], ~~we dropped~~ the 193 users with more than 10,000 connections, arguing that such a large number of connections cannot represent social ties. We argue that this cutoff balances two concerns: it excludes those accounts with so many connections that it brings into question the nature of its connections, and we avoid truncating the tail of the distribution of social connectivity too much, allowing for sociality to range over several orders of magnitude. Many approaches to detect “fake” accounts in social network use the degree of a node as an important input [11].

In Plot A of Figure 6 we plot the sensitivity of fragmentation and diversity to the maximum degree threshold. If we discard all users with more than 100 connections (compared to the 10,000 connection cutoff we use in our paper), fragmentation would be significantly higher and diversity significantly lower than the versions we use in the paper. However this is not a reasonable cutoff as nearly 10% of users have more than 500 connections (see Plot B, Figure 6). The settlement fragmentation and diversity measures are within 5% of the versions we use in the paper if the threshold is set at 500, 1000, or 2000 connections.

In Figure 7 we show the relationship between settlement population and the number of iWiW users listing their location in the settlement, and the share of the population registered to iWiW.

As mentioned in the text, user privacy is a key concern. The anonymized iWiW data was made available to a consortium of researchers in Hungary, each of whom signed a non-disclosure agreement (NDA) to use the data for research purposes only. As a result, only settlement level aggregated data can be shared.

1.2 Corruption risk indicators

In this subsection we go into more detail regarding the individual corruption risk indicators. Each indicator quantifies different ways bureaucrats have excluded competitors in qualitative work on ground truth corruption cases from around the EU [22]. We stress that while no individual indicator or composite measure can credibly suggest that an individual contract was awarded by a corrupt process, aggregated over many contracts issued by the same institution these indicators map highly suggestive patterns. This point is an important motivation for filtering out towns awarding less than five contracts a year.

- Single bidder ($C_{singlebid}$) is an outcome: was the contract awarded in a competition attracting only a single offer.
- Closed procedure ($C_{closedproc}$) indicates when the contracting authority has decided to award a contract by direct negotiation with a firm or via an invitation-only bidding process. This decision can be used to completely subvert competition.
- No call for bids (C_{nocall}) indicates when, in the case that the contract was awarded via an open competition, no contract announcement or call for bids was published in the official procurement journal. A corrupt official can greatly decrease the chance of non-favored firms participating by limiting access to information.
- Long eligibility criteria ($C_{eligcrit}$) captures how bureaucrats can box out specific firms by adding requirements to participation criteria. By including many such restrictions (regarding previous experience, company size, qualifications), a corrupt bureaucrat can systematically exclude non-favored firms.

- Extreme decision period ($C_{decidetime}$) highlights suspicious activity between the end of a competition and the decision to award a contract. If the decision period is extremely short, this suggests that the decision to award a specific firm was premeditated, and that the bids were not carefully checked. If the decision period is very long, it may indicate that legal challenges about the contract may be delaying the award decision.
- Short time to submit bids ($C_{bidtime}$) indicates that favored firms may have been tipped off about a competition for tenders ahead of the public announcement. By leaving only a short time between the announcement and the award for non-favored firms, the corrupt official makes it very difficult to submit a bid. It is important to remember that bids are complex legal documents, including at times cost estimates, schematics, and references.
- Non-price criteria ($C_{nonprice}$) tracks the share of non-price related or subjective criteria in the evaluation of bids. For instance, a corrupt bureaucrat may reject a lower cost bid if, according to a subjective criteria of the quality of a bid, it is less favorably evaluated than that of a higher cost bid of a favored firm.
- Call for bids modified ($C_{callmod}$) checks to see if a call for bids was modified between the initial announcement and the deadline. This potential corruption strategy closely emulates $C_{bidtime}$ in that a corrupt official can suddenly change the specifications or rules of a tender shortly before the deadline.

1.3 Relationship between fragmentation and diversity

Fragmentation and diversity, our measures of bonding and bridging social capital respectively, are positively and significantly correlated ($\rho \approx 0.46$). Though fragmentation considers only edges within the settlement and ego diversity includes external edges, both variables measure modularity in the network. However, according to our hypotheses, they are expected to capture different kinds of socialization. We found that despite their positive correlation these features have opposite relationships with our corruption risk measures: high fragmentation is positively and high diversity is negatively correlated with corruption risk. To test whether inter-settlement edges or the ego focus of diversity does more to distinguish the measure from fragmentation we recalculated the diversity considering only edges within the settlement. This alternative “internal” diversity measure is weakly correlated ($\rho \approx 0.28$) with fragmentation, and strongly correlated with diversity ($\rho \approx 0.72$). This suggests that both the connections to other settlements and the ego-focus of the diversity measure distinguish fragmented settlements from diverse ones.

1.4 Model covariates and controls

In this appendix section we present the settlement-level variables used as controls in our models. We also report their summary statistics. Note that in our models, we scale all features to have mean 0 and standard deviation 1. Our controls mostly refer to data from 2011, when the last large scale Hungarian census took place and the data are of highest quality.

- *Average income per capita (2011)*: Wealthier places tend to be less corrupt [44] as competition for limited resources is expected to create greater incentive to cheat. Data on median income or the income distribution at the settlement level were, to the best of our knowledge, not available in Hungary.
- *Population (log)(2011)*: Larger cities may have different contracting needs, different political and social norms, and different network characteristics.
- *Number of contracts awarded (log)*: Settlements contracting more frequently may be more experienced and may follow better practices. As more people are involved in contracting, corruption may become more difficult.

Statistic	N	Mean	St. Dev.	Min	Max
Closed procedure or single bidder	169	0.59	0.15	0.21	0.92
Average CRI	169	0.28	0.04	0.16	0.40
Fragmentation	169	0.32	0.04	0.16	0.46
Avg. ego diversity	169	0.35	0.07	0.20	0.51
Income per capita (thousands HUF)	169	823.57	189.93	488.44	1,516.55
N contracts (log)	169	4.52	0.69	3.69	6.42
Population (log)	169	9.72	0.89	7.66	12.24
Rate iWiW use	169	0.33	0.06	0.18	0.46
Average mayoral victory margin	169	0.15	0.14	0.00	0.64
% high school graduates	169	47.23	10.22	25.70	76.80
Distance to Budapest (minutes)	169	114.00	54.34	22.55	228.57
Share of population inactive	169	0.30	0.04	0.20	0.40
Unemployment Rate	169	0.06	0.01	0.03	0.09
Share of population 60+	169	0.24	0.03	0.15	0.39
Has university	169	0.25	0.44	0	1

Table 3: Descriptive statistics of key settlement-level variables and controls.

- *Rate of iWiW use (2012)*: The rate of iWiW use both proxies for the economic development of the settlement and controls for differences in observed social network structure resulting from differences in access to the web. Previous work suggests that iWiW users, especially the early adopters, skew young and wealthy [37].
- *Average mayoral victory margin*: Measured across three elections (2002, 2006, 2010), this variable proxies for the lack of political competition in the settlement. The absence of political competition has been shown to correlate with corruption [8].
- *Share of population with at least a high school diploma (2011)*: Education is typically correlated with better control of corruption [59].
- *Share of working-age population inactive and unemployment rate (2011)*: Counting the long-term and short-term unemployed respectively, these variables quantify economic stagnation. The economic hardship connected with high unemployment is conjectured to worsen political corruption [62].
- *The minimum travel distance to Budapest, the capital city*: This variable captures the physical isolation of the settlement from the main economic, political, and social hub of the country. Past research has shown that geographic isolation reduces accountability and increases corruption [10].
- *Share of population over 60 years old (2011)*: This variable controls for the over-representation of the elderly. The elderly are underrepresented on online social networks and tend to use these platforms differently than younger users [52].
- *Whether the settlement has a university (2011)*: This variable controls for the presence of a place of higher education in the settlement, including local branches of universities headquartered elsewhere. This which inflates the number of young people, hence likely iWiW users in the settlement.

1.5 Model results, diagnostics, and feature importances

We present the full model results in Table 4. Note that all variables are standardized with mean 0 and standard deviation 1. This aids interpretation, for example: a one standard deviation increase in the settlement's mayor's average margin of victory increases corruption risk by roughly one quarter of a standard deviation. We also present models including only one of the two network measures in Table 5. The effect and significance of both features is preserved when the other is excluded.

The estimated coefficients of the control variables and their levels of statistical significance offer additional insight into the phenomenon of corruption risk. Wealthier settlements are in general less corrupt, though the effect is not significant for CRI. Rate of iWiW use is not related with corruption risk and this does not change when we include the social capital features. The average mayoral victory margin is a highly significant positive predictor of corruption risk. One potential explanation is that mayors, who do not face significant competition do not fear being voted out of office if they are corrupt. Similarly settlements that are far from Budapest, which our models predict to be significantly more corrupt, may be insulated from investigation by the central authorities simply by being out of the spotlight.

One potential source of bias in the coefficient estimates of multiple regression models is collinearity among the predictors. We test for multi-collinearity for each predictor using a variance inflation factor (VIF) test, defined as the ratio of variance in the full model over the variance of the single-predictor model. We run this diagnostic for each predictor used in models (2) and (4) in the main text and report the results in Table 1.5. A popular rule of thumb is that VIF values under 10 denote acceptable levels of correlation between variables [31]. As it is near our limit, we reran our analyses without the "Share of population inactive" control variable, finding no substantive change in our results. The relevant model tables are available on request.

We show the relative variable importances of Model (6) (column 6 in Table 2), the fully specific model predicting average CRI, using an Analysis of Variance F-test in Figure 8. We include only terms with a significant ANOVA F-test. Though other features have stronger predictive power, the social network features are more useful in predicting corruption risk than economic variables like unemployment, inactivity, and average income.

Dependent variable:	% Closed or single bid.		Average CRI	
	(1)	(2)	(3)	(4)
Fragmentation (Bonding social capital)		0.263*** (0.097)		0.207** (0.092)
Diversity (Bridging social capital)		-0.553*** (0.176)		-0.551*** (0.168)
Income/capita	-0.262 (0.169)	-0.277* (0.162)	-0.075 (0.161)	-0.096 (0.155)
N contracts (log)	-0.313* (0.171)	-0.314* (0.165)	-0.685*** (0.162)	-0.697*** (0.158)
Population (log)	-0.180 (0.143)	0.020 (0.166)	0.118 (0.136)	0.335** (0.159)
Rate iWiW use	0.045 (0.137)	0.037 (0.132)	0.122 (0.130)	0.107 (0.126)
Mayor victory margin	0.278*** (0.089)	0.255*** (0.086)	0.303*** (0.085)	0.281*** (0.082)
% high school grads	0.166 (0.190)	0.374* (0.199)	-0.176 (0.181)	0.040 (0.190)
Distance to Budapest	-0.021 (0.104)	-0.198* (0.112)	0.061 (0.099)	-0.112 (0.107)
Share of pop. inactive	-0.797*** (0.229)	-0.805*** (0.229)	-0.716*** (0.218)	-0.754*** (0.219)
Unemployment Rate	0.239** (0.118)	0.262** (0.113)	0.299*** (0.112)	0.320*** (0.108)
% population 60+	0.501*** (0.163)	0.491*** (0.158)	0.500*** (0.155)	0.503*** (0.151)
Has university	0.351 (0.220)	0.294 (0.221)	0.431** (0.210)	0.352* (0.211)
Constant	1.245* (0.725)	1.206* (0.702)	2.779*** (0.689)	2.790*** (0.671)
Observations	169	169	169	169
Adjusted R ²	0.163	0.230	0.183	0.243
F Statistic	3.967***	4.859***	4.419***	5.142***

Table 4: Settlement-level regression results predicting two corruption risk indicators. For both dependent variables, the first columns (1) and (3) correspond to the base model, predicting corruption risk using only control variables, and the second columns (2) and (4) show results, when the social network features are included. Note that all features are standardized with mean 0 and standard deviation 1. Significance thresholds: *p<0.1; **p<0.05; ***p<0.01.

Dependent variable:	% Closed or single bid.			
	(1)	(2)	(3)	(4)
Fragmentation (Bonding social capital)			0.233** (0.099)	0.263*** (0.097)
Diversity (Bridging social capital)		-0.505*** (0.179)		-0.553*** (0.176)
Income/capita	-0.262 (0.169)	-0.295* (0.166)	-0.243 (0.167)	-0.277* (0.162)
N contracts (log)	-0.313* (0.171)	-0.359** (0.168)	-0.269 (0.169)	-0.314* (0.165)
Population (log)	-0.180 (0.143)	0.083 (0.168)	-0.257* (0.144)	0.020 (0.166)
Rate iWiW use	0.045 (0.137)	0.009 (0.134)	0.073 (0.135)	0.037 (0.132)
Mayor victory margin	0.278*** (0.089)	0.259*** (0.087)	0.276*** (0.088)	0.255*** (0.086)
% high school grads	0.166 (0.190)	0.397* (0.203)	0.126 (0.188)	0.374* (0.199)
Distance to Budapest	-0.021 (0.104)	-0.169 (0.114)	-0.035 (0.102)	-0.198* (0.112)
Share of pop. inactive	-0.797*** (0.229)	-0.931*** (0.229)	-0.675*** (0.232)	-0.805*** (0.229)
Unemployment Rate	0.239** (0.118)	0.253** (0.115)	0.247** (0.116)	0.262** (0.113)
% population 60+	0.501*** (0.163)	0.546*** (0.160)	0.449*** (0.162)	0.491*** (0.158)
Has University	0.351 (0.220)	0.198 (0.222)	0.449** (0.221)	0.294 (0.221)
Constant	1.245* (0.725)	1.426** (0.712)	1.036 (0.720)	1.206* (0.702)
Observations	169	169	169	169
Adjusted R ²	0.163	0.198	0.186	0.230
F Statistic	3.967***	4.460***	4.207***	4.859***

Table 5: Stepwise regressions. The effect and significance of the network features are preserved when including them only one at a time. *p<0.1; **p<0.05; ***p<0.01.

Predictor	VIF
Fragmentation	1.407
Diversity	6.337
Income/capita	5.430
N contracts (log)	3.045
Population (log)	5.892
Rate iWiW use	2.885
Mayor victory margin	1.040
% high school grads	7.106
Share of pop. inactive	9.899
Unemployment Rate	2.360
Distance to Budapest	3.068
% population 60+	5.442
Has university	2.192

Table 6: VIF scores for model predictors.

Figure 6: A) The sensitivity of diversity and fragmentation to changing the maximum degree threshold, relative to the 10,000 degree threshold used in the paper. Error bars represent 95% confidence intervals. The measures are within 5% of the version we use in the paper for cutoffs at or above 500. B) The distribution of user connections on a log scale. Very few users (193) have more than 10,000 connections, while many (405,337) have more than 500.

Figure 7: A) Settlement population and number of iWiW users plotted on a log-log scale. B) iWiW use rate by settlements.

Figure 8: Analysis of Variance F-test feature importances of OLS regression predicting average settlement CRI. We only include significant features, and highlight the network-based social capital measures.

Dependent variable:	% Closed or single bid.		Average CRI	
	(1)	(2)	(3)	(4)
Fragmentation (Bonding social capital)		0.143** (0.069)		0.140** (0.067)
Diversity (Bridging social capital)		-0.358*** (0.138)		-0.440*** (0.134)
Income/capita	-0.324** (0.131)	-0.351*** (0.129)	-0.323** (0.128)	-0.356*** (0.126)
N contracts (log)	-0.389*** (0.118)	-0.384*** (0.118)	-0.669*** (0.116)	-0.672*** (0.115)
Population (log)	-0.064 (0.112)	0.036 (0.131)	0.176 (0.110)	0.318** (0.128)
Rate iWiW use	0.042 (0.094)	-0.001 (0.094)	0.105 (0.092)	0.052 (0.092)
Mayor victory margin	0.176** (0.070)	0.173** (0.069)	0.174** (0.069)	0.169** (0.067)
% high school grads	0.170 (0.122)	0.348** (0.144)	-0.036 (0.120)	0.190 (0.140)
Distance to Budapest	-0.089 (0.078)	-0.204** (0.088)	0.048 (0.077)	-0.093 (0.086)
Share of pop. inactive	-0.456*** (0.138)	-0.440*** (0.138)	-0.430*** (0.135)	-0.422*** (0.134)
Unemployment Rate	0.058 (0.079)	0.064 (0.078)	-0.017 (0.078)	-0.011 (0.076)
% population 60+	0.358*** (0.108)	0.329*** (0.107)	0.283*** (0.106)	0.251** (0.104)
Has University	0.289 (0.204)	0.289 (0.208)	0.406** (0.200)	0.384* (0.202)
Constant	1.561*** (0.463)	1.540*** (0.464)	2.642*** (0.453)	2.652*** (0.451)
Observations	305	305	305	305
Adjusted R ²	0.106	0.129	0.143	0.175
F Statistic	4.271***	4.452***	5.628***	5.974***

Table 7: Settlement-level regression results predicting two corruption risk indicators, including all towns issuing at least one contract a year on average from 2006 to 2014. Note that all features are standardized with mean 0 and standard deviation 1. Significance thresholds: *p<0.1; **p<0.05; ***p<0.01.

Appendix B

Dear Editor,

We write to submit the final revision of our accepted article *Social capital predicts corruption risk in towns* for publication in Royal Society Open Science. We first wish to thank you and the anonymous referees for your helpful suggestions and comments. We have edited the minor corrections on language suggested by Reviewer 2, gratefully acknowledging the thoroughness of his or her reading of our revision. We believe that the manuscript is ready to be published.

Kind regards,

Johannes Wachs, Taha Yasseri, Balazs Lengyel, and Janos Kertesz